

# Parameter dynamics of distributed hydrological model in simulating or forecasting flood processes of urbanizing watersheds

Yangbo Chen[1*], Jun Liu[1], Liming Dong[1]

[1]School of Geography and Planning, Sun Yat-sen University, Guangzhou 510275, China

*Correspondence to:* Yangbo Chen (eescyb@mail.sysu.edu.cn), +86-138-26161264

**Abstract:** In the past decades, the world has experienced rapid urbanization and observed the appearances of large amount urbanizing watersheds with enhanced flooding, which has a constant changing land use/cover(LUC) types as the most significant feature. Simulating and forecasting urbanizing watershed flood processes faces great challenges, one is how to relate model parameters with the changing LUCs to secure an accurately and reliable simulation and forecasting results. In this study, a methodology for simulating and forecasting urbanizing watershed flood processes is proposed, which employs Liuxihe model as the watershed hydrological model. This methodology sets up the Liuxihe model with latest terrain properties, then derives initial parameter look-up table based on terrain properties, and optimizes it if there is observed hydrological data. If there is LUC changes, then the parameters are updated with the changed LUCs based on the optimized parameter look-up tables. Case study in a highly urbanizing watershed in the Pearl River Delta Area in southern China has shown that this method acquires accurate and reliable flood processes simulation results. Further more, this study has proven an assumption that the hydrological model parameters are LUC stationary, i.e., with the LUC changes, the parameter look-up table will not change, parameter look-up table optimized in a specific time with current LUCs will not change even the LUCs changed. With this assumption, the parameter look-up table only needs to be optimized once. This is a science question that has not been not well answered yet by the scientific communities.

**Keywords:** Flood simulation, flood forecasting, land use/cover change, Liuxihe model, parameter optimization



## 1 Introduction

In this study, an urbanizing watershed is referred to a watershed with constant land use/cover (LUC) changing, and has urban land (impervious surface) as its major or significant LUC type. Urbanizing watersheds usually locate in the rapidly developing or urbanizing area. For the past century, the world is in constant urbanization (He et al., 2021; Addae and Dragicevic, 2023), and its urban population reached 50% in 2007 (United Nations, 2014). While for the developing countries, the urbanization trend is still rapidly going (Huang et al., 2022; Xue et al., 2022). For example, intense urbanization has occurred in India and Nigeria from 1970-2010, where 85% and 30% of cropland area within ten km of urban areas converted to urban land respectively (Guneralp et al., 2020). China also experienced a rapid urbanization since 1978, with its urban population reached 55% in 2018 (Zhao et al., 2023; Fang and Wang, 2011). China's urbanization is still going on and quick in some regions (Yu, 2021), with its urban population projected to be 68% in 2050 (Development Research Foundation of China, 2010). During this world urbanization process, lots of urbanizing watersheds appeared worldwide, some of them already have very high urban land percentage over 50% (Wang and Chen, 2019), while others are still in its growing stage. For example, in China's Pearl River Delta Area (PRDA) where the rapidest regional urbanization was observed in China (Li et al., 2011), most of the watersheds have a higher than 30% urban land rate, with some over 50% already (Chen et al., 2015, Zhang et al., 2015).



For an urbanizing watershed, rapid land use/cover (LUC) change, particularly the
converting of vegetated area (pervious surface) into urban land area (impervious
surface), is the most observed direct change caused by human activities. This change
induces increased surface runoff and peak flow, which was well observed and analyzed
(Leopold, 1968; Hollis, 1975; Rose et al., 2001; Yang et al., 2016; Wang et al., 2022;
Zhao et al., 2023). Simulating watershed flood processes has long been the goal of the
world hydrological communities, it is the prerequisite for flood mitigation project
design and flood forecasting. But for an urbanizing watershed, how to consider the
LUCs change in the simulation is still a great challenge.

Watershed hydrological models are the most employed tools for watershed flood
processes simulation and forecast, lumped models are widely used in the early stages
(Refsgaard et al., 1996; Chen et al., 2011), such as the Stanford model (Crawford et al.,
1966), the Xinanjiang model (Zhao, 1977) and the ARNO model (Todini, 1996), only
list a few. Lumped models calibrate model parameters by using series hydrological data
observed in the past, and the LUC change could not be reflected with model structure
or parameter, thus can only be used for simulating the past hydrological processes, not
the changing one. For this reason, lumped hydrological models are not the appropriate
models for simulating or forecasting urbanizing watersheds flood.

Physically based distributed hydrological models (PBDHMs) are the new
development of watershed hydrological models, as the terrain is divided into grid cells
(Freeze and Harlan, 1969; Abbott et al., 1986a, 1986b) and the runoff production and
routing are calculated cell by cell, thus having the potential to better simulate



hydrological processes (Ambroise et al., 1996). Dozens of PBDHMs have been
proposed and widely used in scientific studies, including the SHE model (System
Hydrologue Europeen model) (Abbott et al.,1986a, b), the VIC model (variable
infiltration capacity model) (Liang et al., 1994), the WEP model (Water and Energy
transfer Process model) (Jia et al., 2001) and the Liuxihe model (Chen et al., 2011),
and many others. The most outstanding feature of PBDHMs is that model parameters
have physical meanings, and can be directly derived from the watershed terrain
properties, such as the elevation, soil type and LUC. A table is usually set up to define
the relationship between the parameters and the terrain properties, which is referred to
as the parameter look-up table. This gives PBDHMs potential to simulate or forecast
urbanizing watershed flood as it can relate model parameters with LUC changes.

In the early study, look-up table is proposed based on limited local experiences and
laboratory experiments, which shows big uncertainty and impacts the model's
performance. Recent studies have shown that parameter optimization can improve
PBDHMs performances (Madsen, 2003; Smith et al., 2004; Pokhrel et al., 2012; Chen
et al., 2016), which was assumed previously that parameter optimization is not needed
for PBDHMs. Several methods have been proposed, such as the scalar method (Vieux
et al., 2003; Vieux, 2004) for Vflo model, the SCE algorithm for MIKE SHE (Madsen,
2003), the multi-objective genetic algorithm for WetSpa model (Shafii et al., 2009), the
SCE-UA algorithm (Xu et al., 2012) and the Particle Swam Optimization (PSO)
algorithm (Chen et al., 2016) for Liuxihe model. But these efforts were mainly carried
out in simulating or forecasting flood for natural watershed, i.e., watershed with little
LUC changes.






For simulating or forecasting urbanizing watershed flood processes by using a PBDHM,
big challenges still exist. The biggest one is how to acquire a reliable and accurate look-
up table, so to adjust parameters with changing LUCs. The second is a science question,
i.e., if a reliable and accurate look-up table could be set up, then should it be LUC
stationary? I.e., with the LUC changing, the look-up table should be changed
accordingly or not? There are two purposes for this study, the first is to propose a
methodology for simulating or forecasting urbanizing watershed flood processes by
using a PBDHM with satisfactory model performances, which can relate the model
parameters with the changing LUCs. This methodology employs Liuxihe model as the
PBDHM, and sets up the Liuxihe model with latest terrain properties; then proposes an
initial parameter look-up table based on current parameterization experiences, and
optimizes it if there is observed hydrological data. With parameter look-up table
optimization, the parameter look-up table is reliable and accurate. This methodology
solves the first challeng and has been tested in a highly urbanized watershed in southern
China.

The second purpose is to answer the science question of parameter stationary. The
authors assume that the model parameters are LUC stationary, i.e., with the LUC
changing, the parameter look-up table will not change, i.e., a parameter look-up table
optimized in a specific time with current LUCs will not change after the LUCs changed.
With this assumption, the parameter look-up table only needs to be optimized once.
This assumption has been proven by the simulation results in the case study, so to solve
the second challenge. The remaining parts of this article are organized into 4 sections.
Section 2 introduces the study watershed, the hydrological data, LUC change data and





terrain property data. Section 3 introduces the methodologies, including Liuxihe model
and its parameter look-up table determination. Section 4 introduces the results in the
case study watershed, and section 5 provides a discussion.

## 2 Study watershed and data

### 2.1 Study watershed

Songmushan Watershed(SW) is the upstream of Hanxishui River Basin(HRB) locating
in Dongguan City in southern China. Songmushan Reservoir(SR) was built in the
middle stream of SW, and the watershed area controlled by SR is called as Songmushan
Reservoir Watershed (SRW) in this study. Originating from the Lotus Mountain, SRW
has a drainage area of 54.2 km$^2$, and a river length of 11.19 km. SRW was selected as
the study watershed bacause it is a typical urbanizing watershed in China, and
hydrological data in recent years for this study is also available. Fig. 1 shows the
location of SRW.

Fig. 1 is here


The major topography of SRW is gentle hills. This area enjoys a subtropical monsoon
climate with frequent storms in the summer monsoon season. Average annual
precipitation of SRW is 1674 mm, which caused serious flooding in the past. SRW has
observed rapid urbanization in the past decade, which has created a high percentage of
urbanized land. It is a typical watershed in the Pearl River Delta area that experiences
considerable increasing both in flooding and urbanization.

Songmushan Reservoir was initially built for irrigation in 1958 as Dongguan City was





primarily an agriculture area at that time, and there was no systematical hydrological
observation until 2008. With China's reforming and opening, Dongguan City has been
developing very fast since 1987, and become an international metropolitan. Dongguan
City has been a highly urbanized area since then, and there is almost no agriculture
anymore. For this reason, irrigation is not needed, and the reservoir has the new roles
of flood mitigation and water supply after a heavy flooding in 2006. In 2008, an
automatic hydrological data collecting system was built which includes 6 rain gauges
(Fig. 1) and 1 water level gauge at the dam site.
**2.2 Hydrological data**
In this study, hydrological data of 13 flood events between 2008 and 2015 was obtained,
including precipitation of the 6 rain gauges and the reservoir inflow at one hour interval.
Table 1 shows the basic information of these flood events.

Table 1 is here

**2.3 LUC change data**
Chen et al. (2017) prepared a LUC dataset of the whole Dongguan City using the
Landsat series satellite remote sensing imagery (Irons et al., 2012, Tang et al., 2013). A
total of 12 imageries from 1987 to 2015 at an average 3-year interval were obtained,
and the SVM classifier algorithm (Vapnik, 1995) was employed to estimate the LUCs
accordingly. LUC data at 1987, 1990, 1993, 1996, 1999, 2001, 2003, 2005, 2008, 2011,
2013 and 2015 were prepared for the whole Dongguan City. There are six LUC types,
including urban land (impervious surface), water body, forestry land, farmland,
grassland and bare land. LUCs of SRW in 2008, 2011, 2013 and 2015 were extracted
from this dataset, Fig. 2 shows these results.





Fig. 2 is here


Urban land area of SRW was 18.62% in 2008, but reached 23.40%, 26.24% and 30.37%
in 2011, 2013 and 2015 respectively, this is a significant increasing in urban land under
urbanization, and this watershed could be regarded as an urbanizing watershed as its
LUCs was in constant changing from 2008 to 2015.
**2.4 Terrain property data**
Terrain property data is mainly used for model set up and initial model parameter
deriving, includes DEM, soil type and LUC type. DEM was prepared based on the
topography map surveyed recently by the local government agency, the spatial
resolution is at 30 m grid cell (Fig. 3(a)). The highest, lowest and mean elevation of the
watershed are 489 m, 9 m and 38.1 m respectively.

Soil map was downloaded from the FAO world soil map dataset (www.isric.org) as
shown in Fig. 3(b). There are 4 soil types in the watershed, including urban land, water
body, ferric acrisols and cumulic anthrosols, with areal percentages of 1.0%, 14.0%,
67.0% and 18.0% respectively.

Fig. 3 is here


In this study, a new soil type, the urban land soil type is defined. For which, its land
use/cover is the urban land, but its real soil type beneath the surface could be any type.
In fact, it is a virtual soil type proposed to facilitate the runoff production. So the soil
type data in Fig. 3(b) needs to be adjusted based on this definition. Besides, the urban
land data in Fig. 3(b) was prepared by FAO in 1990, so it is out of date, and have been





updated with the results of Fig. 2 in this study. The final soil types of SRW, adding the
urban land soil type, is produced and shown in Fig. 4, the soil types in 2018, 2011, 2013
and 2015 are different.

Fig. 4 is here

**3 Methodology**
**3.1 Liuxihe model**
The PBDHM employed in this study is the Liuxihe model, which is a physically based,
distributed hydrological model proposed for watershed flood forecasting (Chen, 2009;
Chen et al., 2011; Chen, 2017). But any PBDHMs which could relate its parameters
with LUCs could be employed.

Liuxihe model divides the watershed surface into grid cells, which are categorized as
hill slope cells, river channel cells and reservoir cells. For river channel cells and
reservoir cells, the watershed surface is water, runoff produced in these cells are equal
to the net precipitation. The surfaces of hill slope cells are covered with different land
use/cover (LUC) types, so each hill slope cell has its unique LUC. Currently in Liuxihe
model, there is no urban land LUC type, only vegetated LUCs. Each hill slope cell also
has its own soil type and elevation. LUC type, soil type and elevation are called
watershed terrain properties in Liuxihe model. Runoff is produced first on cells, and
then routed to the watershed outlet via a routing network. Runoff production is
governed by the infiltration, and the soil type is the controlling terrain property for
runoff production. Runoff routing is categorized as hill slope routing, river channel
routing and reservoir routing. The kinematical wave approximation is employed for hill
slope routing, while the diffusive wave approximation for river channel routing.





For Liuxihe model, there is no way to make runoff production and routing calculation
for the urban land grid cells, so in this study, a module that can make this calculation is
added. The urban land surface is impervious, the precipitation falls to this ground
surface is regarded completely converted into surface runoff, and no precipitation is
infiltrated to the soil beneath it. Runoff produced on cells with urban land surface is
equal to precipitation fallen to the surface. The approach used to calculate runoff
production is as below.

$$R_{i,t}=P_{i,t}-E_{i,t} \tag{1}$$

Where $R_{i,t}$, $P_{i,t}$ and $E_{i,t}$ are surface runoff, precipitation and actual evaporation produced
on cell i at time t respectively, and the evaporation could be regarded as water surface
evaporation if there is surface runoff, otherwise it is zero.

As only the hill slope cell may have urban land surface, so the runoff routing on urban
land cell is hill slope routing. In Liuxihe model, hill slope routing is solved by using
kinematic wave approximation. For hill slope routing, the governing factors are the
slope of the cell and the roughness coefficient of the surface. For the hill slope routing
on urban land surface, the same approach is used but using different roughness
coefficient. The above approaches for runoff production and routing on urban land cell
has been developed and embedded into the currently used Liuxihe model software tool.
**3.2 Liuxihe model parameter look-up table determination**
Liuxihe model is a distributed hydrological model, so each grid cell has its own
parameters, i.e., 13 parameters (Chen et. al, 2011). The parameters in each grid cell are
divided into 4 categories, including climate-based parameters, topography-based
parameters, vegetation-based parameters and soil-based parameters (Chen et. al, 2016).
The parameters' values are related to only one category terrain property of its grid cell,


i. e., climate-based parameters are only related to the climate condition, the topography-
based parameters are only related to the topography, vegetation-based parameters are
only related to the land use/cover types, and the soil-based parameters are only related
to the soil types. There is only one climate-related parameter, i.e., the reference
evaporation which is regarded as the same for all grid cells. There are two topography-
based parameters, including flow directions and slopes for hill slope cells and river
channel cells. There are also two vegetation-based parameters, the evaporation
coefficient and roughness. There are 8 soil-based parameters, including soil property
coefficient, soil thickness, hydraulic conductivity under saturated condition, soil water
contents under saturated condition, field condition, and wilting condition. There is one
parameter for underground water routing which is regarded as the same for all grid cells,
and is also a soil-based parameter.

Liuxihe model takes two steps to determine model parameters, firstly deriving initial
parameter look-up tables from the watershed terrain property data, and then optimizing
them. For a specific watershed studied, Liuxihe model first proposes parameter look-
up tables, which are two-dimensional tables referring the values of parameters with the
terrain properties, for example, with soil type Ferric Acrisols, the parameter value of
soil water content under saturated conditions is referred to as 46.1%. Based on these
parameter look-up tables, the parameters of each grid cell could be determined
according to the grid cell's terrain properties, including DEM, LUCs and soil types. As
climate-based parameters take the same value for all grid cell, so there is no need for a
look-up table for the climate-based parameters. While the topography-based parameters
are calculated directly based on the DEM using the D8 method (O'Callaghan et al.,
1984; Jensen et al., 1988), so there is no need for a look-up table for the topography-




based parameters also. Therefor there are two parameter look-up tables, one is for
vegetation-based parameters, and another is for soil-based parameters.

Liuxihe model proposed ways for determining the two parameter look-up tables. For
the vegetation-based parameters look-up table, the referring values are decided from
laboratory experiments and local experiences, or even from references or results from
other watersheds. There are two vegetation-based parameters, the evaporation
coefficient and roughness. For the soil-based parameters look-up table, Liuxihe model
employs the Soil Water Characteristics Hydraulic Properties Calculator (Arya et al.,
1981) to calculate the referring values based on the soil texture, organic matter, gravel
content, salinity and compaction. With these parameters look-up table, based on its
terrain properties, the initial parameters for each grid cell could be derived. With this
way, if the terrain properties of each grid cell are available, then the initial parameters
could be proposed.

As the initial parameters derived with the above method are highly experience-based,
and current parameterization experiences are very limited, so the initial parameters have
uncertainty, thus model performance could not be secured. To improve model
performance, Liuxihe model optimizes the initial parameters by using optimization
algorithm, this is the second step of Liuxihe model parameters determination. From
past experiences of Liuxihe model parameterization, it has been found that parameter
optimization could largely improve the model's performance. Besides, in optimizing
model parameters, hydrological data from only one flood events is enough, not like
lumped hydrological mode, series of hydrological data is requirred. This is very
important for an urbanizing watershed as it usually has limited hydrological data, no

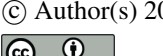



long series of hydrological data.

Currently two algorithms have been proposed for Liuxihe model parameter
optimization, one is SCE-UA algorithm (Xu et al., 2012), another is Particle Swam
Optimization (PSO) algorithm (Chen et al., 2016). In optimizing parameters, Liuxihe
model does not optimize all parameters of each grid cells, but optimizes the parameters
look-up tables. I.e., an adjusting coefficient for each terrain property is proposed, so the
optimized variables are limited, which makes the calculation practical, otherwise, even
with the fastest computers in the world, the optimization is not feasible.
**3.3 Dynamic parameter updating and parameter stationary**
For an urbanizing watershed, the terrain properties, particularly the LUCs are in
constant changes, so after parameter look-up table is optimized based on hydrological
data from one specific flood event and terrian properties at a specific date, model
parameters should be updated if the terrain properties changes, it is called in this study,
dynamic parameter updating, it is also the core concept of this study, that the model
parameters are dynamically changing with terrain property changes. Only with this
dynamic parameter updating, the model performance can be secured. But this parameter
updating is based on the assumption that the parameter look-up tables are LUCs
stationary, i.e., the look-up tables are not changed with the terrain property changing.
Otherwise, the updated parameters could not improve the model performance, or the
parameter look-up table needs to be optimized again with the changed LUCs and new
observed hydrological data.

Do the parameter look-up tables change with the watershed terrain property changing,
it is a science question that has not been answered and fully studied by the scientific



communities. The authors assume that the parameter look-up tables of Liuxihe model
are LUC stationary, and the simulation results in the case study will validate this
assumption.

## 4 Results

### 4.1 Liuxihe model set up

The DEM produced in this study with spatial resolution of 30 m is used to divide the
studied watershed into 62942 grid cells, which are further divided into 658 river cells,
53435 hill slope cells and 8849 reservoir cells, based on the method employed in
Liuxihe model. A 3-order river network is derived using the D8 method (O'Callaghan
et al., 1984; Jensen et al., 1988) and Strahler river ordering method (Strahler, 1957)
based on the DEM. The river network is further divided into 24 virtual sections based
on 4 virtual nodes. In the Liuxihe model, the virtual river cross section shape is
assumed trapezoidal, and the river size is estimated based on satellite remote sensing
images. The structure of the Liuxihe model for SRW set up in this study is shown in
Fig. 5. The time resolution of the Liuxihe model set up in SRW is 1 hour, the same
with that of the observed hydrological data. Precipitation from rain gauges is
interpolated to the grid cells by using the Thiesson Polygon method (Thiessen, 1911).

Fig. 5 is here

Flow directions and slopes are derived using the D8 method (O'Callaghan et al., 1984,
Jensen et al., 1988) based on the DEM. The climate-based parameter, i.e., the potential
evaporation is estimated as 5 mm/day for each grid cell according to daily evaporation
observations in this region. According to previous studies of Liuxihe model





parameterization and references (Chen et al., 1995; Zhang et al., 2006, 2007; Guo et al.,
2010; Li et al., 2013), the initial look-up table for vegetation-based parameters is
proposed and listed in Table 2.

Table 2 is here


Based on past modeling studies (Zaradny, 1993; Anderson et al., 1996; Shen et al., 2007;
Zhang et al., 2015), the soil water content under wilting conditions takes 30% of the
soil water content under field conditions, and the soil porosity coefficient takes the
value of 2.5. Based on local experiences, the estimated soil layer thickness is listed in
Table 3. The Soil Water Characteristics Hydraulic Properties Calculator proposed by
Arya et al. (1981) is employed to calculate the soil water contents under saturation
condition and field condition, and the hydraulic conductivity under saturation
conditions, as listed in Table 3.

Table 3 is here


For grid cells with urban land soil type, all the soil-based parameters are set to zero.
This reflects the hydrological response of urban land soil type, i.e., the precipitation
falling onto urban land will be converted into surface runoff completely, no
precipitation will be infiltrated to the soil or stored on the surface.

For Liuxihe model, hydrological data from only one flood event is needed for parameter
optimization, and Particle Swam Optimization(PSO) is the official optimization
algorithm, which has been tested and proven to be effective. In this study, hydrological





data from flood event 20080625 is used for parameter optimization, and PSO algorithm
is employed to optimize the parameters, while LUCs in 2008 is used. The optimized
parameter look-up tables are called parameter-20080625-2008 to distinguish
parameters optimized with different hydrological data and LUCs in different year. In
parameter-20080625-2008, the first number is the flood event number with its
hydrological data being used for parameter optimization, while the second one is the
year of the LUCs which is used in the parameter optimization. I.e., Parameter-
20080625-2008 is the optimized parameters by using hydrological data from flood
event 20080625 and LUCs in 2008. Fig. 6 shows the evolution results of parameters,
adaptive values and evaluation indices during the parameter optimization process.

Fig. 6 is here


With 9 evolution, the model parameters approached their optimal values, and the
simulated hydrograph with optimized parameters fits the observed flood event well as
shown in Fig 6(d), this means the PSO algorithm has good performance for Liuxihe
model parameter optimization.

From the result of Fig 6(a), it has been found that the initial value of soil property
coefficient is quite different from its optimized value, but with the optimization of PSO
algorithm, its optimized value is obtained, this implies that the PSO algorithm has good
convergence, and well suits Liuxihe model parameter optimization.
**4.2 Flood simulation with Parameter-20080625-2008**
With the above optimized Parameter-20080625-2008, the other 12 flood events were
simulated, while in this sumulation, the LUCs in 2018 are used for all the 12 flood


events, that means the parameters are regarded not changed during the watereshed
urbanization, and the model parameters are not updated dynamically with the LUC
changing. Four evaluation indices, including Nash-Sutcliffe coefficient, mean relative
error, peak flow error and peak flow timing error, has been calculated and listed in Table
4, the simulated hydrographs are shown in Fig. 7.

Table 4 is here


Fig. 7 is here


From the results shown in Table 4 and Fig. 7, it has been found that for all the 12 flood
events, the simulated hydrographs are similar with the observations in shape. In average
for all the 12 flood events, the Nash-Sutcliffe coefficient is 0.79, the mean relative error
is 63.91%, the peak flow error is 19.47%, while the peak flow timing error is -0.58 hour.
From these results, the flood processes of SRW have been simulated reasonable by
Liuxihe model set up in this study.

From the above results, we also find that the four evaluation indices get worse with
time goes. For example, the average peak flow error for flood events in 2008 and 2009
is 6.7%, 35.88% in 2011, 17.15% in 2013 and 2014, and 27.87% in 2015, in general,
the average peak flow error gets bigger as time goes. For the Nash-Sutcliffe coefficient,
those in 2008 and 2009, in 2011, in 2013 and 2014, and in 2015 are 0.835, 0.775, 0.753
and 0.76, a similar trend with peak flow error. Based on these results, it can be proposed
that the model parameters should have changed with time going, i.e., with LUC changes,
and the model parameters need to be adjusted with the changing LUCs. To verify this



opnion, the dynamical parameter updating is tested in the follow-up section.

**4.3 Flood simulation with dynamic updating to parameter-20080625-2008**
Based on the dynamic parameter updating method proposed in this study, parameters
used for simulating flood events in 2011, in 2013 and 2014, in 2015 are updated with
the LUCs in 2011, 2013 and 2015 respectively based on parameter-20080625-2008.
The dynamically updated model parameters in 2011, 2013 and 2015 are different from
each other, so are from parameter-20080625-2008, which is called parameter-
20080625-2008-updated. With these parameters, 8 flood events(parameters for the
flood events in 2008 and 2009 are not updated) are simulated again, the four evaluation
indices have been calculated and listed in Table 5, the simulated hydrographs are shown
in Fig. 7 also to make comparison with those results simulated with no parameter
updating.

Table 5 is here


From the results shown in Fig. 7 and Tabel 5, it has been found that the model
performance has been improved with dynamic parameter updating. For example, for all
the 8 flood events, the simulated hydrographs fit those of the observations better than
those simulated with no parameter updating. The Nash-Sutcliffe coefficients of all the
simulated flood events with updated parameters gets higher, except those of flood event
no. 20150520 and 20150720. The average Nash-Sutcliffe coefficient increasing is 0.764,
a 0.3% incrasing. While for the peak flow error, all flood events have observed
decreasing, the average decreasing is 66.81%, a very significant model performance
imporvement. These results imply that with dynamical parameter updating, Liuxihe





model has a much better performance in simulating the flood events of SRW, i.e., model
parameters are in dynamic changing with the LUC changing, and dynamical parameter
updating with the LUC changing is needed. This confirms the dynamic characteristics
of model parameters.

**5 Discussions**
**5.1 Effect of parameter optimization on model performance**
To test the effectiveness of parameter optimization, the 12 flood events are simulated
with the initial parameters, and the results are shown in Fig 8, to make comparision, the
simulated hydrographs with dynamically updated parameters are also shown. From the
results it could be found that the simulation results with initial and dynamically updated
parameters are quite different. Though both the simulated hydrographs have similar
patterns, but the flows simulated with the initial parameters are generally much lower
than the observations, and those simulated with the dynamically updated parameters
well fit the observation.

Fig. 8 is here


The four evaluation indices of the 12 flood events is calculated and listed in Table 6.
Compared with the simulation with initial parameter, the simulation with dynamically
updated parameters has been improved much based on these evaluation indices. Among
them, the average Nash-Sutcliffe coefficient increased 68.2%, correlation coefficient
increased 3.2%, peak flow error reduced 86.4%, water balance coefficient increased
45.8%. These results show that parameter optimization is needed and feasible even for
distributed hydrological model.




Table 6 is here


**5.2 Parameter stationary**
In above sections, the dynamic parameter updating was based on the optimized
parameter look-up tables with LUCs in 2018. There appears a question, should the look-
up table be optimized with the latest LUCs, not the LUCs at a specific time? I.e., is the
look-up table no-stationary to LUCs during urbanization. If yes, then the look-up table
needs to be optimized with the latest LUCs, and done again when there is significant
LUC change. Otherwise, the look-up table can be optimized with LUCs at any time,
and there is no need to optimize it very often. To answer this question, in this study, the
parameters were optimized with LUCs in 2011, 2013 and 2015 also, and the
hydrological data used for parameter optimization were from flood events 20110516,
20130815 and 20150520 respectively, these parameters are called parameter-
20110516-2011, parameter-20130815-2013, and parameter-20150520-2015
respectively. Then the parameters are dynamically updated with latest LUCs, and are
called parameter-20110516-2011-updated and parameter-20130815-2013-updated
respectively, there is not parameter-20150520-2015-updated. I. e., dynamical parameter
updating is time forward, not time backward. For example, parameter-20110516-2011-
updated only update parameters with LUCs in 2013 and 2015, not in 2008 and 2011;
parameter-20130815-2013-updated only update parameters with LUCs in 2015, not in
2008, 2011 and 2013, so there is not parameter-20150520-2015-updated. The
dynamically optimized and updated parameters are then employed to simulate the flood
events, and the results are shown in Fig. 9. The four evaluation indices are calculated
and listed in table 7.




Table 7 is here

Fig. 9 is here


Both the simulated flood hydrographs with and without dynamic parameters optimizing
and updating have no obvious differences, so based on the above methods and results,
it can be concluded that the parameters are stationary during the urbanization, i.e.,
during the LUC changing period. There is no need to optimize the look-up table very
often with rapid LUC changing, parameter optimization and updating is most important.
**5.3 Impact of LUC changes on flood responses**
Based on the above results, it could be found that with the LUC changes, the flood
response changes also. To quantitatively analysis this effect, the peak flow and urban
land area rate of flood events from 2011 to 2015 are extracted from the above results
and listed in Table 8. The values with no-update are the simulated values with
parameter-20080625-2008, while the ones with update are the simulated values with
parameter-20080625-2008-update.

Table 8 is here

From the results, it could be found that from 2008 to 2011, the SRW observed an urban
land rate change from 18.62% to 23.4%, a 25.67% increasing. For flood event
20110516, with the same precipitaion, the peak flow will change from 87.08 m$^3$/s to
99.42 m$^3$/s, having a 14.2% increasing. While for flood event 201100808, the peak flow
change is from 103.68 m$^3$/s to 117.21 m$^3$/s, having a 13.1% increasing. Both these
events are light flood, the peak flow increasing has similar magnitude.





But from 2008 to 2013, the SRW observed an urban land rate increasing of 40.92%.
For flood event 20130815, which is regarded as a heavy flood event, the peak flow
increasing is 9.0%, while for flood event 20140511, the peak flow increasing is 12.8%,
for flood event 20140819, this is 14.6%. The latter two flood events are regarded as
medium. With these results, it can be concluded that the much heavier of the flood
magnitude, the more increasing of peak flow.

From 2008 to 2015, the SRW observed an urban land rate increasing of 63.10%. For
flood event 20150520, which is regarded as a light flood event, and flood events
20150523 and 20150720, which are regarded as a medium flood event, the peak flow
increasing are 56.3%, 18.5%, and 12.2% respectively. This implies that for the light
flood event, the peak flow increases much more. Based on this analysis, with the
increasing of the urban land area rate, the peak flow of a flood event will increase, and
the light flood event has the most peak flow increasing, while the heavy one has the
least peak flow increasing.

**6 Conclusions**
In this study, a method is proposed for accurately simulating flood processes of
urbanizing watersheds that appear during the world urbanization process, which
employs the Liuxihe model, a physically based distributed hydrological model as the
flood simulation tool. This method first derives initial parameter look-up tables, and
then optimizes it, and dynamically updates the parameter with the changing LUCs to
improve the model performance. A case study has been carried out in the Songmushan
Reservoir Watershed, a highly urbanizing watershed in the Pearl River Delta Area in
southern China which experienced rapid urbanization in the past decade. Based on the
results, following conclusions have been proposed.




1. The methodology proposed in this study could be used for simulating and forcasting
urbanizing watershed flood processes with good model performance.

2. For an urbanizing watershed, terrain properties are in changing, and model
parameters are in changing also due to terrain properties changing, this is called
model parameter dynamics. Model parameters should be updated with the LUC
changes.

3. Parameter look-up table of physicall based distributed hydrological model is LUC
stationary, i.e., the parameter look-up table only needs to be determined once during
the watershed urbanization.

4. With same precipitation, flood peak flow will increase due to urban land rate
increases. The much heavier the precipitation, the less increasing of the peak flow.

5. Parameter optimization is effective and needed in controlling parameter uncertainty
for physically based distributed hydrological model.

**Competing interests:** The contact author has declared that none of the authors
has any competing interests.





**Acknowledgements:** This study was supported by the National Natural Science
Foundation of China (NSFC) (no. 51961125206), and the Science and Technology
Program of Guangdong Province (no. 2020B1515120079).



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





## Figures

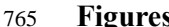

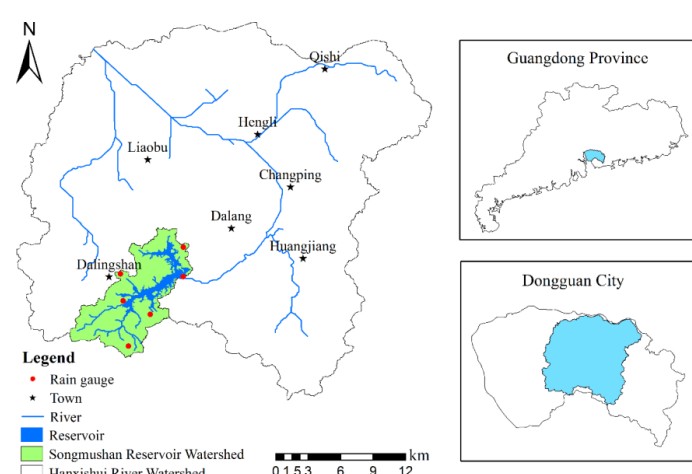

Fig. 1 Location of Songmushan Reservoir Watershed(SRW)

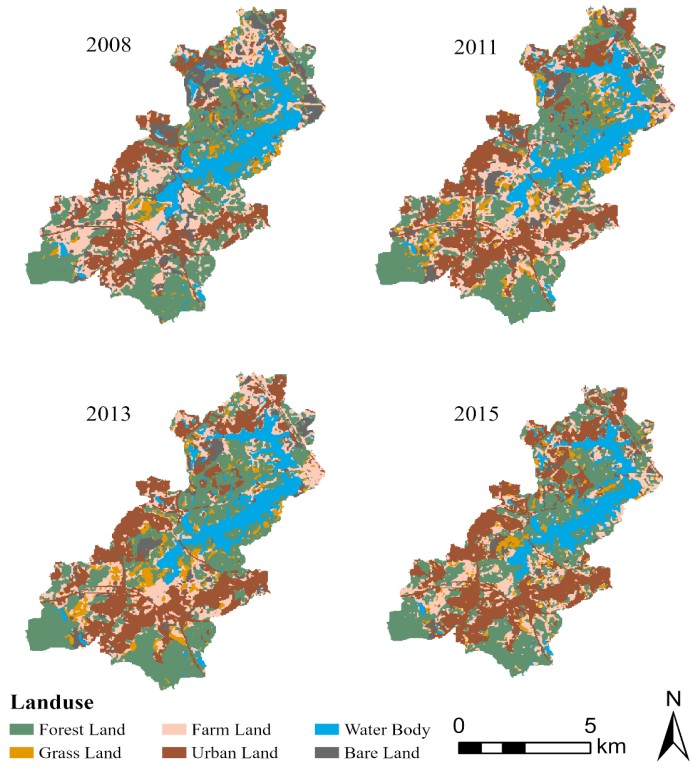

Fig. 2 LUCs of SRW






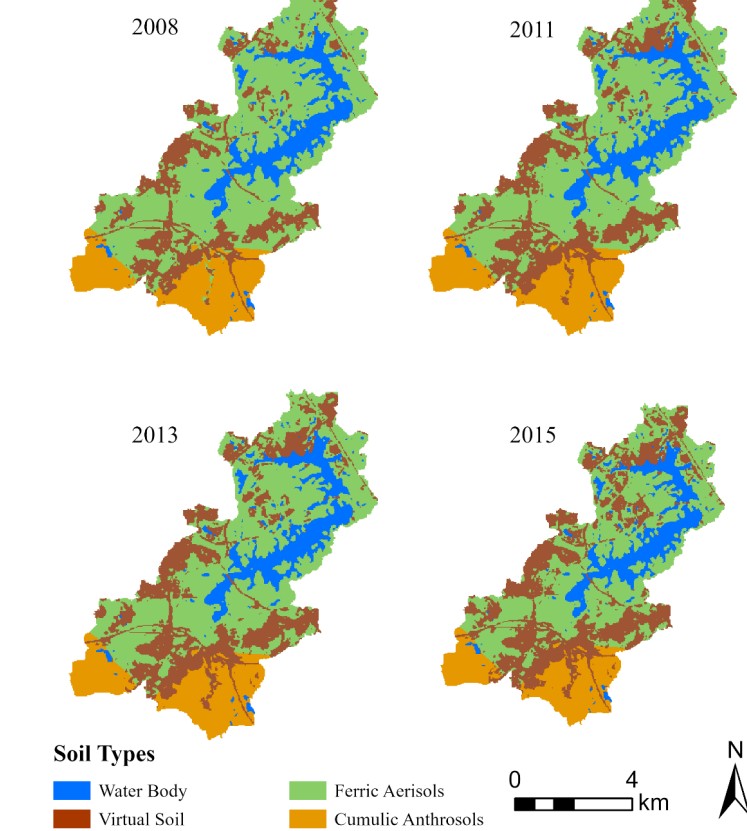

(a) DEM                        (b)Soil type

Fig. 3 Terrain property of SRW


2008                    2011

2013                    2015

**Soil Types**

| | | | |
|---|---|---|---|
| ■ Water Body | ■ Ferric Aerisols | | |
| ■ Virtual Soil | ■ Cumulic Anthrosols | | |


Fig. 4 Soil types of SRW adding urban land soil






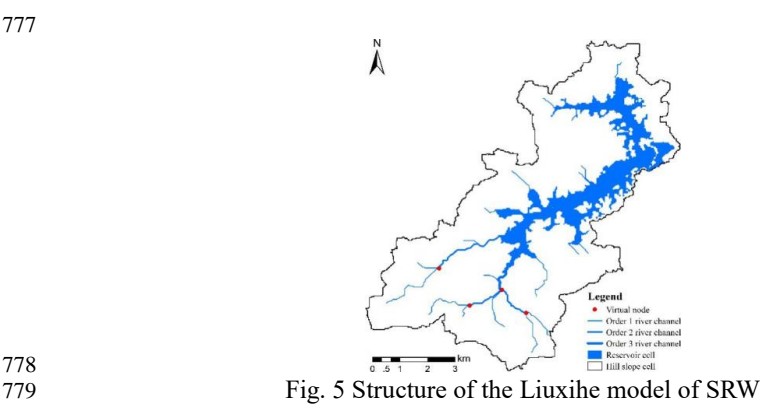

Fig. 5 Structure of the Liuxihe model of SRW


(a) The evolution of parameters     (b)The evolution of objective function

(c)The evolution of statistical indicators        (d)Simulation(20080625)

Fig. 6 results of parameter optimization




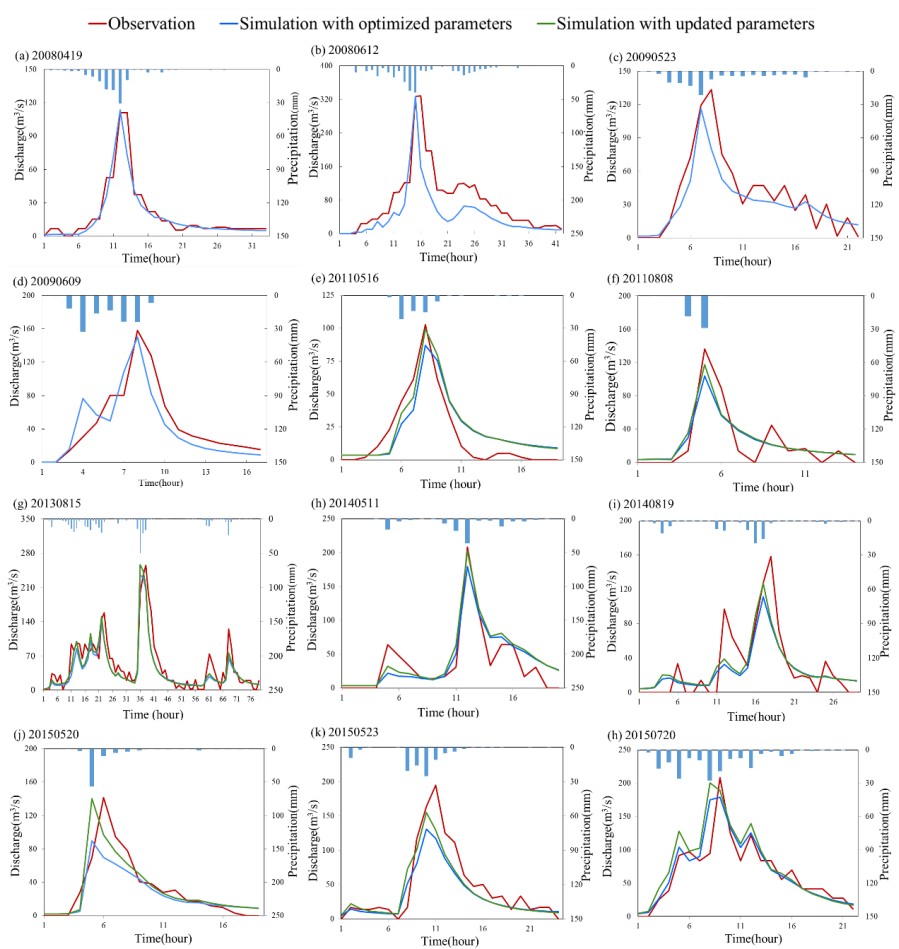


Fig. 7 Simulated flood hydrographs with Parameter-20080625-2008 and Parameter-

20080625-2008-updated



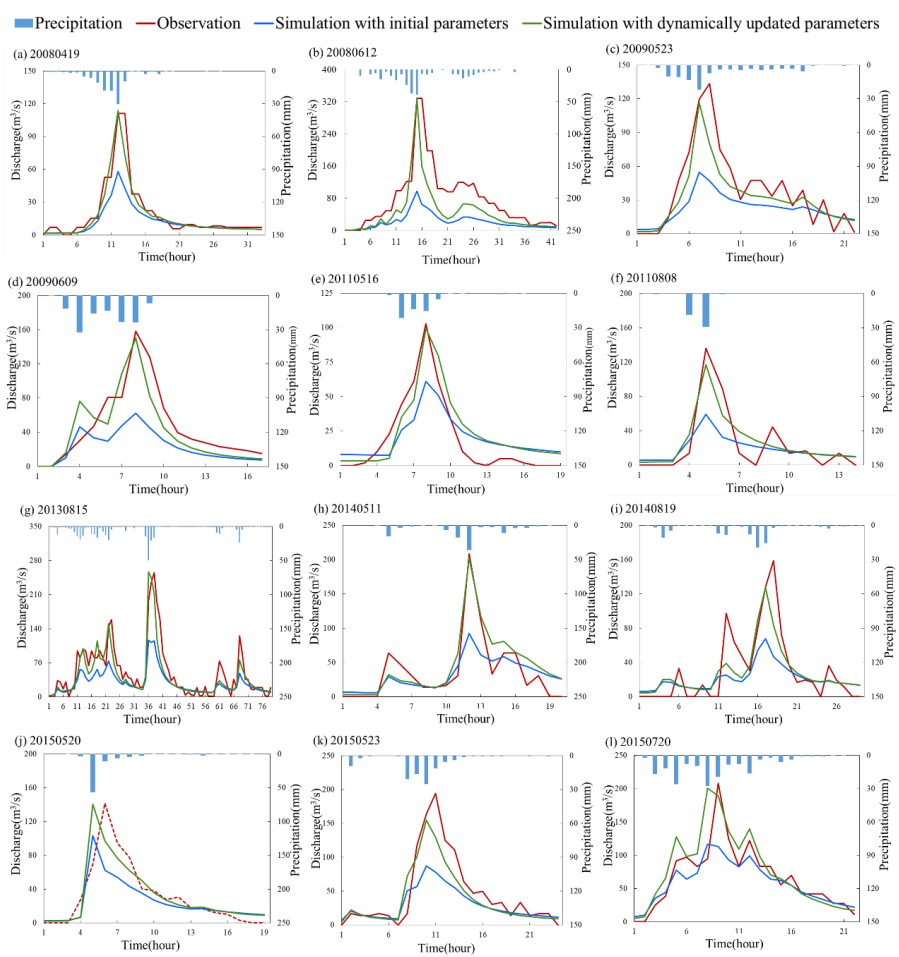


Fig. 8 Simulated flood hydrographs with initial parameter and dynamically updated

parameters




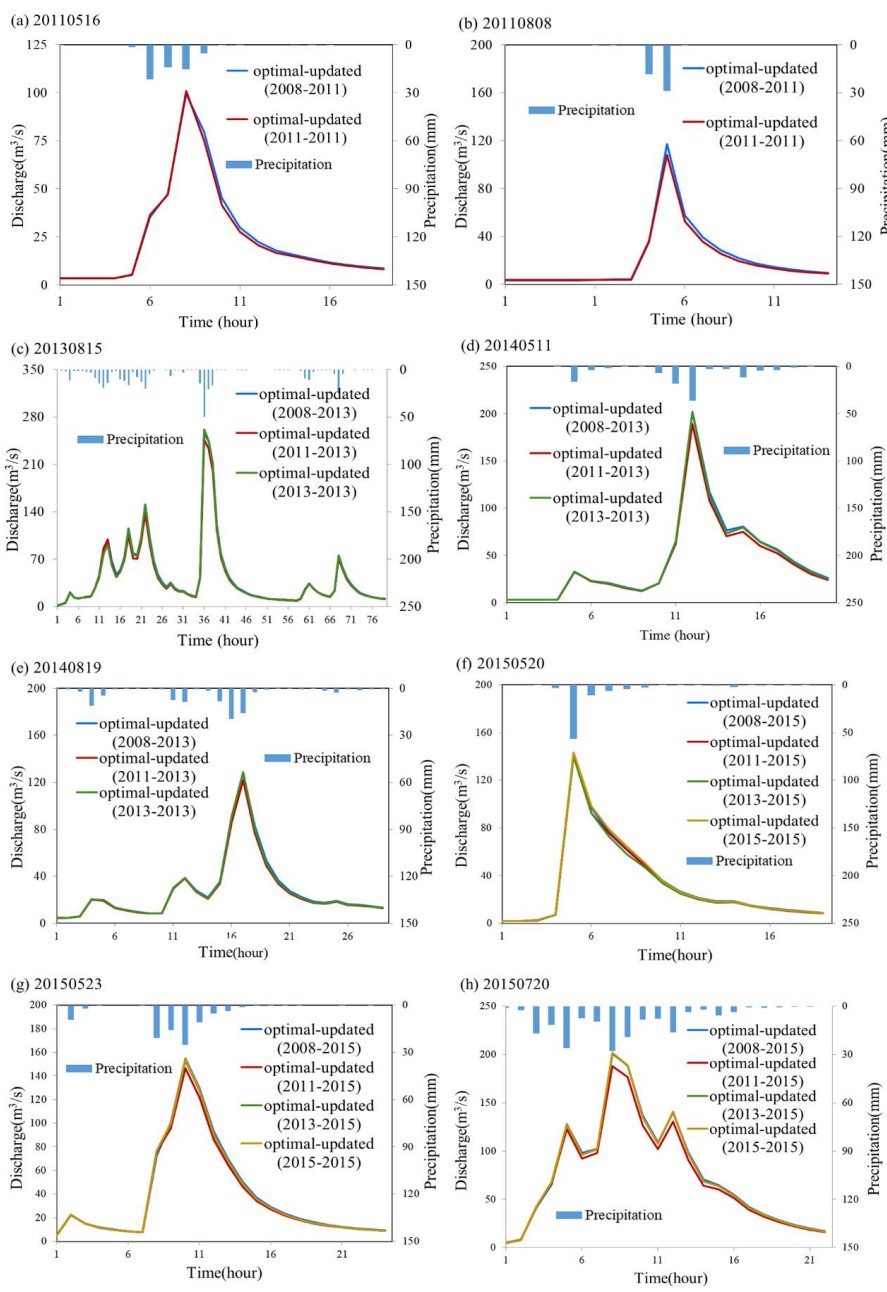


Fig. 9 Simulated flood hydrographs with dynamic parameter optimizing and updating.
* optimal-update (2008-2011) represents simulation results based on the 2008 LUC
optimization parameters and the 2011 LUC updated parameters.





## Tables

Table 1 Brief information of flood events

| Flood event no. | Start time (yyyymmddhh) | End time (yyyymmddhh) | Total rainfall (mm) | Peak flow (m³/s) | Flood magnitude* |
|---|---|---|---|---|---|
| 20080419 | 2008041910 | 2008042018 | 110.6 | 111.1 | light |
| 20080612 | 2008061219 | 2008061412 | 271.0 | 328.6 | heavy |
| 20080625 | 2008062500 | 2008062623 | 360.3 | 445.8 | heavy |
| 20090523 | 2009052304 | 2009052401 | 104.4 | 133.3 | light |
| 20090609 | 2009060900 | 2009060916 | 127.7 | 158.3 | medium |
| 20110516 | 2011051608 | 2011051702 | 60.1 | 102.8 | light |
| 20110808 | 2011080811 | 2011080900 | 48.6 | 136.1 | light |
| 20130815 | 2013081517 | 2013081823 | 351.3 | 254.7 | heavy |
| 20140511 | 2014051103 | 2014051122 | 110.7 | 208.3 | medium |
| 20140819 | 2014081914 | 2014082018 | 98.0 | 158.3 | medium |
| 20150520 | 2015052009 | 2015052103 | 90.1 | 141.7 | light |
| 20150523 | 2015052305 | 2015052404 | 100.9 | 194.4 | medium |
| 20150720 | 2015072022 | 2015072119 | 171.8 | 208.3 | medium |

\* Flood magnitude is a qualitative measurement to the flood based on the peak flow of a flood event. I.e., for a flood event, if its peak flow is below 150 m³/s, then it's flood magnitude is light. On the other hand, if its peak flow is over 250 m³/s, it's flood magnitude is heavy. For other flood events, it's medium.

Table 2 Initial look-up table for vegetation-based parameters

| **Vegetation** | Evaporation coefficient | Roughness |
|---|---|---|
| Forestry land | 0.7 | 0.55 |
| Grassland | 0.6 | 0.18 |
| Urban land | 1.0 | 0.01 |
| Bare land | 0.4 | 0.12 |
| Farmland | 0.55 | 0.36 |





Table 3 Initial look-up table for soil-based parameters

| Soil type | Soil water content under saturated conditions (%) | Soil water content under field capacity conditions (%) | Soil hydraulic conductivity under saturated conditions(mm·h$^{-1}$) | Soil layer thickness (mm) |
|---|---|---|---|---|
| Urban land | 0 | 0 | 0 | 0 |
| Ferric Acrisols | 46.1 | 26.5 | 20.78 | 1500 |
| Cumulic Anthrosols | 45.8 | 35.3 | 2.81 | 850 |


Table 4 Evaluation indices of simulated flood events with Parameter-20080625-2008

| Flood event no. | Nash-Sutcliffe coefficient | Mean relative error (%) | Peak flow error (%) | Peak flow timing error/hour |
|---|---|---|---|---|
| 20080419 | 0.93 | 31.29 | 0.49 | 0 |
| 20080612 | 0.70 | 38.83 | 0.94 | -1 |
| 20090523 | 0.81 | 144.10 | 23.96 | -1 |
| 20090609 | 0.90 | 24.60 | 1.41 | 0 |
| 20110516 | 0.77 | 189.10 | 15.27 | 0 |
| 20110808 | 0.78 | 52.65 | 56.48 | 0 |
| 20130815 | 0.80 | 97.98 | 7.71 | -2 |
| 20140511 | 0.80 | 48.54 | 13.99 | 0 |
| 20140819 | 0.66 | 36.06 | 29.75 | -1 |
| 20150520 | 0.69 | 43.34 | 36.58 | -1 |
| 20150523 | 0.78 | 39.52 | 32.85 | -1 |
| 20150720 | 0.81 | 20.87 | 14.19 | 0 |
| average | 0.79 | 63.91 | 19.47 | -0.58 |







Table 5 Evaluation indices of simulated flood events with parameter-20080625-2008-
updated

| Flood event no. | Parameter | Nash-Sutcliffe coefficient | Correlation coefficient | Relative error (%) | Peak flow error (%) | Water balance coefficient | Peak flow timing error/hour |
|---|---|---|---|---|---|---|---|
| 20110516 | No-updated | 0.769 | 0.889 | 189.10 | 15.30 | 1.17 | 0 |
| | Updated | 0.812 | 0.922 | 186.70 | 3.30 | 1.27 | 0 |
| | Difference | 0.043 | 0.033 | -2.40 | -12.00 | 0.10 | 0 |
| | Increase(%) | 5.600 | 3.700 | 1.27 | -78.43 | 8.40 | 0 |
| 20110808 | No-updated | 0.784 | 0.919 | 57.30 | 23.80 | 1.01 | 0 |
| | Updated | 0.806 | 0.914 | 62.70 | 13.90 | 1.08 | 0 |
| | Difference | 0.022 | -0.005 | 5.40 | -9.90 | 0.07 | 0 |
| | Increase(%) | 2.800 | -0.500 | 9.42 | -41.60 | 7.10 | 0 |
| 20130815 | No-updated | 0.802 | 0.921 | 97.90 | 7.70 | 0.78 | -2 |
| | Updated | 0.813 | 0.915 | 93.40 | 0.60 | 0.84 | -2 |
| | Difference | 0.011 | -0.006 | -4.50 | -7.10 | 0.05 | 0 |
| | Increase(%) | 1.400 | -0.700 | 4.60 | -92.21 | 6.90 | 0 |
| 20140511 | No-updated | 0.804 | 0.900 | 48.50 | 13.90 | 1.08 | 0 |
| | Updated | 0.819 | 0.921 | 52.40 | 3.00 | 1.20 | 0 |
| | Difference | 0.015 | 0.021 | 3.90 | -10.90 | 0.11 | 0 |
| | Increase(%) | 1.900 | 2.300 | 8.04 | -78.42 | 10.40 | 0 |
| 20140819 | No-updated | 0.659 | 0.865 | 36.10 | 29.80 | 0.83 | -1 |
| | Updated | 0.689 | 0.849 | 33.60 | 19.50 | 0.92 | -1 |
| | Difference | 0.030 | -0.016 | -2.50 | -10.30 | 0.09 | 0 |
| | Increase(%) | 4.600 | -1.800 | 6.93 | -34.56 | 10.50 | 0 |
| 20150520 | No-updated | 0.691 | 0.873 | 43.30 | 36.60 | 0.80 | -1 |
| | Updated | 0.683 | 0.837 | 46.20 | 0.90 | 1.01 | -1 |
| | Difference | -0.008 | -0.036 | 2.90 | -35.70 | 0.21 | 0 |
| | Increase(%) | -1.200 | -4.100 | 6.70 | -97.54 | 26.40 | 0 |
| 20150523 | No-updated | 0.782 | 0.959 | 39.50 | 32.80 | 0.73 | -1 |
| | Updated | 0.823 | 0.932 | 42.60 | 20.40 | 0.83 | -1 |
| | Difference | 0.041 | -0.027 | 3.10 | -12.40 | 0.09 | 0 |
| | Increase(%) | 5.200 | -2.800 | 7.85 | -37.80 | 12.70 | 0 |
| 20150720 | No-updated | 0.810 | 0.914 | 20.90 | 14.20 | 1.05 | 0 |
| | Updated | 0.666 | 0.889 | 29.70 | 3.70 | 1.15 | -1 |
| | Difference | -0.144 | -0.025 | 8.80 | -10.50 | 0.10 | -1 |
| | Increase(%) | -17.800 | -2.700 | 42.11 | -73.94 | 9.70 | 0 |
| average | No-updated | 0.763 | 0.905 | 66.58 | 21.76 | 0.93 | -0.63 |
| | Updated | 0.764 | 0.897 | 68.41 | 8.16 | 1.03 | -0.75 |
| | Difference | 0.001 | -0.008 | 1.84 | -13.60 | 0.10 | -0.13 |
| | Increase(%) | 0.313 | -0.825 | 10.86 | -66.81 | 11.51 | 20.00 |







Table 6 The evaluation indices of simulated flood events with initial and dynamically

updated parameters

| Flood event no. | Parameter | Nash-Sutcliffe coefficient | Mean relative error (%) | Peak flow error (%) | Peak flow timing error/hour |
|---|---|---|---|---|---|
| 20080419 | initial | 0.604 | 0.310 | 0.507 | 0 |
| | updated | 0.928 | 0.313 | 0.005 | 0 |
| | difference(%) | 53.6 | 1.0 | -99.0 | 0.0 |
| 20080612 | initial | 0.010 | 0.559 | 0.708 | -1 |
| | updated | 0.701 | 0.388 | 0.009 | -1 |
| | difference(%) | 6910.0 | -30.6 | -98.7 | 0.0 |
| 20090523 | initial | 0.304 | 1.489 | 0.628 | -1 |
| | updated | 0.809 | 1.441 | 0.239 | -1 |
| | difference(%) | 166.1 | -3.2 | -61.9 | 0.0 |
| 20090609 | initial | 0.335 | 0.385 | 0.611 | 0 |
| | updated | 0.897 | 0.246 | 0.014 | 0 |
| | difference(%) | 167.8 | -36.1 | -97.7 | 0.0 |
| 20110516 | initial | 0.636 | 1.831 | 0.481 | 0 |
| | updated | 0.812 | 1.867 | 0.033 | 0 |
| | difference(%) | 27.7 | 2.0 | -93.1 | 0.0 |
| 20110808 | initial | 0.482 | 0.526 | 0.568 | 0 |
| | updated | 0.806 | 0.627 | 0.139 | 0 |
| | difference(%) | 67.2 | 19.2 | -75.5 | 0.0 |
| 20130815 | initial | 0.476 | 1.038 | 0.543 | -2 |
| | updated | 0.813 | 0.934 | 0.006 | -2 |
| | difference(%) | 70.8 | -10.0 | -98.9 | 0.0 |
| 20140511 | initial | 0.548 | 0.432 | 0.556 | 0 |
| | updated | 0.819 | 0.524 | 0.030 | 0 |
| | difference(%) | 49.5 | 21.3 | -94.6 | 0.0 |
| 20140819 | initial | 0.398 | 0.412 | 0.573 | -1 |
| | updated | 0.689 | 0.336 | 0.195 | -1 |
| | difference(%) | 73.1 | -18.4 | -66.0 | 0.0 |
| 20150520 | initial | 0.576 | 0.513 | 0.269 | -1 |
| | updated | 0.683 | 0.462 | 0.009 | -1 |
| | difference(%) | 18.6 | -9.9 | -96.7 | 0.0 |
| 20150523 | initial | 0.518 | 0.413 | 0.550 | -1 |
| | updated | 0.823 | 0.426 | 0.204 | -1 |
| | difference(%) | 58.9 | 3.1 | -62.9 | 0.0 |
| 20150720 | initial | 0.724 | 0.222 | 0.439 | -1 |
| | updated | 0.666 | 0.297 | 0.037 | -1 |
| | difference(%) | -8.0 | 33.8 | -91.6 | 0.0 |
| average | initial | 0.468 | 0.678 | 0.536 | -0.667 |
| | updated | 0.787 | 0.655 | 0.077 | -0.667 |
| | difference(%) | 68.2 | -2.3 | -86.4 | 0.0 |







Table 7 Evaluation indices of simulated flood events with dynamic parameter optimizing and updating

| Flood event no. | Optimizing year | Nash-Sutcliffe coefficient | Correlation coefficient | Relative error (%) | Peak flow error (%) | Water balance coefficient | Peak flow timing error/hour |
|---|---|---|---|---|---|---|---|
| 20110516 | 2008 | 0.812 | 0.922 | 1.867 | 0.033 | 1.269 | 0 |
| | 2011 | 0.845 | 0.934 | 1.724 | 0.019 | 1.231 | 0 |
| 20110808 | 2008 | 0.806 | 0.914 | 0.627 | 0.139 | 1.076 | 0 |
| | 2011 | 0.780 | 0.909 | 0.621 | 0.206 | 0.997 | 0 |
| 20130815 | 2008 | 0.813 | 0.915 | 0.934 | 0.006 | 0.838 | -2 |
| | 2011 | 0.789 | 0.909 | 0.962 | 0.032 | 0.799 | -2 |
| | 2013 | 0.797 | 0.908 | 0.922 | 0.028 | 0.822 | -2 |
| 20140511 | 2008 | 0.819 | 0.921 | 0.524 | 0.030 | 1.195 | 0 |
| | 2011 | 0.839 | 0.923 | 0.487 | 0.091 | 1.127 | 0 |
| | 2013 | 0.828 | 0.923 | 0.512 | 0.031 | 1.181 | 0 |
| 20140819 | 2008 | 0.689 | 0.849 | 0.336 | 0.195 | 0.918 | -1 |
| | 2011 | 0.662 | 0.844 | 0.329 | 0.231 | 0.871 | -1 |
| | 2013 | 0.675 | 0.841 | 0.332 | 0.186 | 0.904 | -1 |
| 20150520 | 2008 | 0.683 | 0.837 | 0.462 | 0.009 | 1.006 | -1 |
| | 2011 | 0.663 | 0.825 | 0.451 | 0.006 | 0.979 | -1 |
| | 2013 | 0.658 | 0.821 | 0.455 | 0.009 | 0.966 | -1 |
| | 2015 | 0.677 | 0.837 | 0.455 | 0.011 | 1.018 | -1 |
| 20150523 | 2008 | 0.823 | 0.932 | 0.426 | 0.204 | 0.826 | -1 |
| | 2011 | 0.776 | 0.918 | 0.465 | 0.246 | 0.784 | -1 |
| | 2013 | 0.803 | 0.923 | 0.452 | 0.211 | 0.814 | -1 |
| | 2015 | 0.806 | 0.923 | 0.449 | 0.207 | 0.816 | -1 |
| 20150720 | 2008 | 0.666 | 0.889 | 0.297 | 0.037 | 1.151 | -1 |
| | 2011 | 0.721 | 0.883 | 0.288 | 0.097 | 1.084 | -1 |
| | 2013 | 0.659 | 0.885 | 0.304 | 0.033 | 1.144 | -1 |
| | 2015 | 0.657 | 0.885 | 0.304 | 0.032 | 1.145 | -1 |

Table 8 impact of urbanization on peak flow

| Flood event no. | Peak flow (m³/s) | | | Urbanization rate (%)* | | |
|---|---|---|---|---|---|---|
| | no-update | update | increase(%) | no-update | update | increase(%) |
| 20110516 | 87.08 | 99.42 | 14.2 | 18.62 | 23.4 | 25.67 |
| 20110808 | 103.68 | 117.21 | 13.1 | 18.62 | 23.4 | 25.67 |
| 20130815 | 235.08 | 256.19 | 9.0 | 18.62 | 26.24 | 40.92 |
| 20140511 | 179.18 | 202.07 | 12.8 | 18.62 | 26.24 | 40.92 |
| 20140819 | 111.23 | 127.43 | 14.6 | 18.62 | 26.24 | 40.92 |
| 20150520 | 89.85 | 140.44 | 56.3 | 18.62 | 30.37 | 63.10 |
| 20150523 | 130.58 | 154.72 | 18.5 | 18.62 | 30.37 | 63.10 |
| 20150720 | 178.78 | 200.59 | 12.2 | 18.62 | 30.37 | 63.10 |

*Urbanization rate is the rate of urban land area to the whole watershed area.