# Peer review of "Parameter dynamics of distributed hydrological model in simulating or forecasting flood processes of urbanizing watersheds"

_Hydrology and Earth System Sciences, 2023_

## Referee Comment (RC2)

To whom it may concern,

Thank you for giving me the opportunity to review your paper.

From my understanding, the main aspect of this paper is "Dynamic Parameter Updating." The method uses different parameter values over time based on the changing land cover. The parameter values are updated from the same look-up table at each flood event based on the land cover present at the event.

I recommend rejection as I do not believe the methods shown are novel enough to solicit publication in HESS. I believe that comparing Particle Swarm with other algorithms has already been done many times. I recommend going beyond a simple case study to better prove your lookup values are applicable to other urban catchments, or by adding more algorithms to compare against.

I wish you the best of luck moving forward,

Tadd Bindas

**Major Comments:**
1. Can the figures be added in line with more descriptive captions? This seems minor but can deter readers from wanting to read your content if they cannot find your figures easily. I had to open two tabs when reviewing this, one for figures and one for the text, as I didn't want to lose my spot when reading.
2. Line 282-283: Is there a citation for the referring look-up values?
3. Lines 293-296. Can you better describe the initial values and the process that the Liuxihe model uses to attain them? This would benefit the reader as you refer to these metrics heavily in your paper, say they are, "experience-based," but don't provide a method to calculate them/reproduce them.
4. Line 377: Is there a citation on why PSO is effective with getting parameters and is "proven effective" when only using one flood event? Given the small amount of flood events used or parameter calibration, there is a chance that parameters are overfit
5. Line 430, 436: Can there be a simpler flood naming convention? Using a long name with little formatting, parameter-20080625-2008, seems complex. Perhaps calling it

Parameter set A, and then have a look-up table that maps your set letter to your dates. Using parameter-20080625-2008-updated could be called Parameter set B.

6.  The following comments are for Figure 6:
    a.  For panel (a), what is the adjusting factor?
    b.  For panel (b), what is your objective function? This is not described in the paper.
    c.  For panel (c ), it looks like your evaluation metrics (R, NSE, etc) start extremely close to their final values (after evolution 50)
    d.  For panel (d), Can you provide a simulation with initial parameters as well? Based on the slight changes in metrics over all 50 epochs, my hypothesis is the initial parameters hydrograph would look very similar since your metrics in panel (c ) didn't vary much. A suggestion would be to compare the percent error of the initial simulation vs observations and the optimal parameter simulation vs observations. If your routing timing, and flood matching is more stable (closer to 0) after PSO, you can state that your optimal solutions are of a good fit.
    e.  Line 391-394: You stated that because there was good fitting, the PSO algorithm worked. Can you add a quantitative metric/comparison with initial values to better show this?
7.  Line 461: Are the initial parameters the ones used in Section 4.1, or the parameters generated from PSO at parameter-20080625-2008?
8.  Figure 8: Can this figure be reworked? The text is very small/un-readable in the middle figures, and the precipitation varying with time change is hard to see.
9.  For the conclusions, I'm not convinced by Conclusions 3, and 5. There was no experiment outlined for how the PSO algorithm was run, and if the data was overfit or not (only results were shown in Figure 6). Furthermore, the initial lookup table is based on "current experiences," which are not mentioned in this paper.

**Minor Comments:**
1.  Line 163: The number 6 should be written out as a word, six.
2.  Figures 2, 4: Is there a better way to highlight the LUC change without having four similar basin plots? I can't really tell the difference between any of the yearly changes.
3.  Line 306, 376: You already defined PSO, no need to redefine it.
4.  Line 376: There needs to be a space inserted between Optimization and the (
5.  Line 376: It's *Particle Swarm,* not Particle Swam. This needs to be corrected in a few places.
6.  Line 376: What algorithm/coding package was used for PSO?

7. Line 436: the number 8 should be written out as a word, eight.

8. Line 436: There needs to be a space inserted between events and the (

9. Line 564: spelling error: *physically*

---

## Author Comment (AC1)

RC1: 'Comment on hess-2023-233'

General comment:

The paper has its value and represents a fair contribution to the scientific community mostly towards the discussion of the needed for adopting dynamic parameter optimisation in hydrological models by answering a so-far not completely answered scientific question. However there are many aspects of the manuscripts that ought to be corrected/included before the paper may pass through a peer-review process. After the completion of these corrections I will consider the paper for peer-review in HESS and personally I believe it would fit the journal.

General Reply:

Dear Reviewer, thank you very much for your efforts to review our paper, and gave us a positive comment. We have revised our paper based on your and another reviewer's comments, and a point-to-point response to your comments have been presented below. Please also note that the revised manuscript is attached in this end.

R1-C1. The paper requires substantial gramatical revisions. The structure is therefore confusing to the readers and need a proper revision in terms of the appropriate use of the English language.

AC1: We are very sorry about the English, please forgive us as we are not native English speakers. We have made substantial gramatical revisions, and have it proofreaded by a native speaker before it was sumbitted. We made further improvement once again, and hope the revised version is better to your satisfaction.

R1-C2.  1. INTRODUCTION: The paper aims mainly to answer an interesting research question regarding the need or not of optimising hydrological parameters after significant LUC changes at catchment level. I miss a proper cover of the current literature about the subject in the introduction. Remember that this part is needed to prepare the reader to what is gonna be mainly discussed in the paper. Therefore, the authors need to cover recent publications in the subject. Other studies where they considered the optimisation and got similar results? or not? Unless there is no literature about this topic (which I doubt) the authors should cover this more properly.

AC2: Thanks. During our study, we made a full literature review, and did not find literature covering the issue about parameter dynamics, the parameter updating, and parameter stationary. Most of the literatures derive model parameters from the terrain properties they obtained directly, and no parameter optimization in most cases. This is the main reason we hava made this study. Thank you for your reminder, we reveiwed the literatures again, and did not find new literature on this topic. So we are not able to expend the introduction.

We think this is partly because this kind of studies needs employing distributed hrologcial model and huge hydrologicial data to simulate hydrological processes under difference LUCs conditions, it is always not easy and very time-consuming. We spent huge amount of times in more than 3 years to finish this study, it is really not easy. We fully understand that why there is not so much study in this topic.

R1-C3. 2.2. HYDROLOGICAL DATA: Remember to always cite the sources of the dataset used. Where is the source of the streamflow used? Is there an official government website? Where readers can obtain the same dataset in order to reproduce the present work?

AC3: Thank you for your reminder, I fully agree with you. But as the hydrological data used in this study is not from a public source, and we have promised to use this data only for this study (Hydrological data, like in many other countries, is not public data in China), so we are not able to publicize the data. But we add the data source in the paper: the administration of Songmushan Reservoir, and hope you understand our situation.

R1-C4. The methodology is confusing. You need to provide a proper and enjoyable workflow guiding the readers to what you did. what were your hypothesis and what they might expect. Think that your readers might want to reproduce this work in a coherent work and then write the most direct, and at the same time sufficient, as possible. Consider the use of flowcharts or framework figures to help the reader to understand the work-flow.

AC4: Thank you for your suggestion, very good suggestion. We rewrite the whole section 3, add a sub-section "3.1 Overview of the methodology", and a technical roadmap figure is presented to ilustrate the methodology concisely, followed by detailed description to the important methods. As it is not allowed to submit the revised manuscript at this moment, so we attach the whole section 3 in the end with revision marks.

R1-C5 3.3. Dynamic parameter updating and parameter stationary: This entire section is very confusing, The authors do not make clear what they actually did here. Please consider restructuring this section since it is one of the most important of the manuscript.

AC5: Thank you very much. We rewrite the whole section 3. Please see the revised one in this end.

R1-C6. 5 Discussions: This part should still be part of the results and not of the discussion.

AC6: Thank you very much for your comment. We think this part can be partly results, and partly discussion also, so we merged the two parts as one part called results and discussions. As it is not allowed to submit the revised manuscript at this moment, so we attach the whole section 4 in the end with revision marks.

R1-C7. The authors need to include a proper discussion in the paper. The current discussion part is actually part of the results and cannot be considered a discussion. For the discussion the authors need to explore the current literature and how close/different are the results found by other authors to the results presented here. This paper conclusions have a big potential for the subject, but to be properly effective they need to be as much as possible discussed in view of what has been done/ is being done by others.

AC7: Thanks. As mentioned above, there is no this kind of study, so we are not able to compare something as you suggested. We think this part can be partly results, and partly discussion also, so we merged the two parts as one part called results and discussions.

**Attachment: revised section 3 and section 4**

type data in Fig. 3(b) needs to be adjusted based on this definition. Besides, the urban land data in Fig. 3(b) was prepared by FAO in 1990,  it is out of date, and was updated with the results of Fig. 2 in this study. The final soil types of SRW, inclusive of the newly introduced urban land soil type, are illustrated in Fig. 4. Notably, the soil types for the years 2018, 2011, 2013, and 2015 exhibit variations

.

Fig. 4 is here

## 3 Methodology

### 3.1 Overview of the methodology

The methodology proposed in this study has three major steps, including preparation, model set up, and flood simulation, which are briefly introduced below, and a technical roadmap depicting the whole precedure is presented in Figure 5. For some important methods, they are explained in more details in the following sub-sections.

Figure 5 is here

1. Preparation

In the preparation step, two things need to be done. Firstly, a physically based, distributed hydrologic model needs to be selected as the flood processes simulation tool, and the one selected for this study needs to be able to simulate the flood processes as accurate as possible in the urbanizing watersheds, and to be able to relate its parameters with the LUCs changes. As mentioned in the introduction section, there are already many PBDHMs, and Liuxihe model is selected as the PBDHM in this study, which will be introduced in more detail in the following sub-section. But some other PBDHMs may also be employed as the hydrological model if they satisfy the above requirements.

Secondly, a watershed needs to be selected as the study case which should be an urbanizing watershed, i.e., significant LUC changes should be observed in the study period. There should be hydrological data observation during this period, data for model set up and flood processes simulation should also be available. As introduced in section

2, the Songmushan Watershed has been selected as the study watershed, which has appropriate data for this study, it is an ideal study case.

2. Model set up

The second step is to set up the selected hydrological model in the selected watershed, which includes several jobs. Firstly, to set up the model structure with available terrain property data. Different from lumped hydrological model, which has the same model structure in different watersheds, PBDHMs have different model sturctures in different watershed, which could be set up by using terrain property data, including DEM, soil types, LUCs, different PBDHMs have different model structures also.

Secondly, the initial model parameters need to be determined, usually PBDHMs derive their initial model parameters from the terrain properties, different model usually has its own ways to do this job. For Liuxihe model, the parameterization method is introduced in the following sub-section.

Thirdly, to optimize model parameters. As mentioned in the introduction section, initial model parameters ususlly have high uncertainty, and parameter optimization is an effective way to control this uncertainty. For Liuxihe model, it has proposed effective parameter optimization methodology, which will be introduced in more details in the following sub-section.

3.Flood simulation

The last step is to simulate the flood events observed in the study watershed during the LUCs changing period, and based on these simulation results with different LUCs and parameter combination scenarios, conclusions could be proposed. In this study, three kinds of simulations need to be done. Firstly, the flood simulation with optimized model parameters, so to prove the model employed in this study is rational, and can simulate the urbanizing watershed flood processes effective. While at the same time, to prove that parameter optimization is needed even for PBDHMs, which could improve the model performance, and is feasible computationally.

The second simulation is about the parameter's dynamics and parameter updating method proposed in this study. This could be done by simulating flood events with and without parameter updating, then comparing the two simulation results to find the conclusion on parameter dynamics and parameter updating. This simulation is described in more details in the following sub-section.

The third simulation is about parameters LUCs stationary. These simulations simulate flood processes with dynamic parameter optimization and updating, and with only parameter updating. By comparing these results, if parameters are LUCs stationary can be proposed. These simulations are described in more details in the following sub-section.

**3. 2 Liuxihe model and structures**

[revised manuscript text omitted]

**3.3 4 PDynamic parameter dynamics and parameter updating and parameter**

**stationary**

In this study, parameter dynamics is defined as when LUCs change, the model
parameters should change also with the LUCs changing. This is an assumption
proposed by the authors. Further, if the parameter dynamics assumption is correct, then
the model parameters should be updated with the changing LUCs. As fFor an
urbanizing watershed, the terrain properties, particularly the LUCs are in constant
changes, so after parameter look-up table is optimized based on hydrological data from
one specific flood event and terrian properties at a specific date, model parameters
should be updated if the terrain properties changes based on the parameter dynamics
assumption. I, it is called in this study, dynamic parameter updating in this study, thisit
is also the core concept of this study, that the model parameters are dynamically
changing with terrain property changes. Only with this dynamic parameter updating,
can the model performance can be secured.

It seems the parameter dynamics assumption is obviously correct based on human's direct response or experiences, but we must prove this assumption scientifically. In this study, we use the simulation results to validate this assumption. We first simulate flood processes of observed flood events with the optimized hydrological model parameters, and also simulate the same flood events with updated model parameters, then compare these simulated flood processes. If both the simulated flood processes are the same, it implies that the model parameters do not change with the LUCs change. If there is significant difference between these two results, it implies that the model parameters changed due to LUCs change. Further, if the simulated flood processes with updated parameters fits the observation better, then it can be concluded that parameter updating is necessity, and model performance can be improved with parameter updating. In this study, the above assumption will be validated in the case study.

But this parameter updating is based on the assumption that the parameter look-up tables are LUCs stationary, i.e., the look-up tables are not changed with the terrain property changing. Otherwise, the updated parameters could not improve the model performance, or the parameter look-up table needs to be optimized again with the changed LUCs and new observed hydrological data.

**3.5 Parameter stationary**

But thisThe  parameter updating method proposed in this study is based on the aassumption that the parameter look-up tables are LUCs stationary, i.e., the look-up tables aredo not changed with the terrain property changing, including LUCs change, otherwise, the parameter look-up tables need to be optimized again after there is significant LUCs change.  So, if parameters are LUCs stationary, then there is no need to optimize the parameter look-up tables very often, Ootherwise, they need to be optimized again and again. The authors assume that the parameters are

LUCs stationary, which will be proved in this study.

Parameters LUCs stationary is a science question that has not been answered and seldom studied by the world scientific communities. In this study, our approach to address this question involves simulating flood processes and conducting comparisons.

We introduce a methodology termed "dynamic parameter optimization and updating."

This process entails optimizing the parameter look-up tables when significant LUCs changes occur and subsequently updating model parameters with the newly optimized parameter look-up tables. Then flood processes are simulated under two conditions:

dynamic parameter optimization and updating, and solely parameter updating. In the latter condition, the parameter look-up tables remain unaltered post-initial optimization and persist unchanged despite substantial land use/cover (LUC) modifications.

Comparing these two results, if the simulation results are almost the same or very close, it can be inferred that the parameter look-up tables remain consistent despite significant

LUC changes. This inference supports the assumption that the model parameters are

LUC stationarity

otherwise, we can not validate this assumption.

**4 Results and discussions**

**4.1 Liuxihe model set up**

The DEM produced in this study with spatial resolution of 30 m is used to divide the studied watershed into 62942 grid cells, which are further divided into 658 river cells, 53435 hill slope cells and 8849 reservoir cells, based on the method employed in Liuxihe model. A 3-order river network is derived using the D8 method (O'Callaghan et al., 1984; Jensen et al., 1988) and Strahler river ordering method (Strahler, 1957) based on the DEM. The river network is further divided into 24 virtual sections based on 4 virtual nodes. In the Liuxihe model, the virtual river cross section shape is assumed trapezoidal, and the river size is estimated based on satellite remote sensing images. The structure of the Liuxihe model for SRW set up in this study is shown in Fig. 56. The time resolution of the Liuxihe model set up in SRW is 1 hour, the same with that of the observed hydrological data. Precipitation from rain gauges is interpolated to the grid cells by using the Thiesson Polygon method (Thiessen, 1911).

Fig. 5 6 is here

Flow directions and slopes are derived using the D8 method (O'Callaghan et al., 1984, Jensen et al., 1988) based on the DEM. The climate-based parameter, i.e., the potential evaporation is estimated as 5 mm/day for each grid cell according to daily evaporation observations in this region. According to previous studies of Liuxihe model parameterization and references (Chen et al., 1995; Zhang et al., 2006, 2007; Guo et al., 2010; Li et al., 2013), the initial look-up table for vegetation-based parameters is proposed and listed in Table 2.

Table 2 is here

Based on past modeling studies (Zaradny, 1993; Anderson et al., 1996; Shen et al., 2007;

Zhang et al., 2015), the soil water content under wilting conditions takes 30% of the soil water content under field conditions, and the soil porosity coefficient takes the value of 2.5. Based on local experiences, the estimated soil layer thickness is listed in

Table 3. The Soil Water Characteristics Hydraulic Properties Calculator proposed by

Arya et al. (1981) is employed to calculate the soil water contents under saturation condition and field condition, and the hydraulic conductivity under saturation conditions, as listed in Table 3.

Table 3 is here

For grid cells with urban land soil type, all the soil-based parameters are set to zero.

This reflects the hydrological response of urban land soil type, i.e., the precipitation falling onto urban land will be converted into surface runoff completely, no precipitation will be infiltrated to the soil or stored on the surface.

For Liuxihe model, hydrological data from only one flood event is needed for parameter optimization, and Particle Swarm Optimization(PSO) is the official optimization algorithm, which has been tested and proven to be effective. In this study, hydrological data from flood event 20080625 is used for parameter optimization, and PSO algorithm is employed to optimize the parameters, while LUCs in 2008 is used. The optimized parameter look-up tables are called parameter-20080625-2008 to distinguish parameters optimized with different hydrological data and LUCs in different year. In parameter-20080625-2008, the first number is the flood event number with its hydrological data being used for parameter optimization, while the second one is the year of the LUCs which is used in the parameter optimization. I.e., Parameter-20080625-2008 is the optimized parameters by using hydrological data from flood event 20080625 and LUCs in 2008. Fig. 6 7 shows the evolution results of parameters, adaptive values and evaluation indices during the parameter optimization process.

Fig. 6 7 is here

With 9 evolution, the model parameters approached their optimal values, and the simulated hydrograph with optimized parameters fits the observed flood event well as shown in Fig 67(d), this means the PSO algorithm has good performance for Liuxihe model parameter optimization.

From the result of Fig 67(a), it has been found that the initial value of soil prorosityperty coefficient is quite different from its optimized value, but with the optimization of PSO algorithm, its optimized value is obtained, this implies that the PSO algorithm has good convergence even the initial values is far from its optimal one, and well suits Liuxihe model parameter optimization. From these resutls, it also found that Liuxihe model well suits the flood simulation of urbanized watershed, and could relate its parameters with the LUCs, and the parameters could be optimized with the initial values.

**4.2 Flood simulation for parameter dynamics and dynamic updating**

1. Flood simulation with Parameter-20080625-2008

Using the above optimized parameter-20080625-2008, simulations were conducted for the remaining 12 flood events. Notably, in this simulation, the LUCs data for the year

批注 [KY1]: 之前的句子有点长, 且逗号连接了太多句子, 好像不太符合英文书面表达

2018 was employed consistently across all 12 flood events. This approach assumes that the parameters remain constant throughout the watershed's urbanization process, indicating that the model parameters are not dynamically updated in response to changing LUC conditions. With the above optimized Parameter 20080625 2008, the other 12 flood events were simulated, while in this sumulation, the LUCs in 2018 are used for all the 12 flood events, that means the parameters are regarded not changed during the watereshed urbanization, and the model parameters are not updated dynamically with the LUC changing. Four evaluation indices, including Nash-Sutcliffe coefficient, mean relative error, peak flow error and peak flow timing error, has been calculated and listed in Table 4, the simulated hydrographs are shown in Fig. 78.

                          Table 4 is here

                          Fig. 7 8 is here

From the results shown in Table 4 and Fig. 78, it has been found that for all the 12 flood events, the simulated hydrographs are similar with the observations in shape. In average for all the 12 flood events, the Nash-Sutcliffe coefficient is 0.79, the mean relative error is 63.91%, the peak flow error is 19.47%, while the peak flow timing error is -0.58 hour.

From these results, the flood processes of SRW have been simulated reasonablreasonablye by Liuxihe model set up in this study.

From the above results, we also find that the four evaluation indices get worse with time goes. For example, the average peak flow error for flood events in 2008 and 2009

is 6.7%, 35.88% in 2011, 17.15% in 2013 and 2014, and 27.87% in 2015, in general, the average peak flow error gets bigger as time goes. For the Nash-Sutcliffe coefficient, those in 2008 and 2009, in 2011, in 2013 and 2014, and in 2015 are 0.835, 0.775, 0.753

and 0.76, a similar trend with peak flow error. Based on these results, it can be proposed that the model parameters should have changed with time going, i.e., with LUCs changes, and the model parameters need to be adjusted with the changing LUCs. To verify this opinion, the dynamical parameter updating is tested in the follow-up sectioning paragraph.

4.24.3 Flood simulation with dynamic updating to parameter-20080625-2008

Based on the dynamic parameter updating method proposed in this study, parameters used for simulating flood events in 2011, in 2013 and 2014, in 2015 are updated with the LUCs in 2011, 2013 and 2015 respectively based on parameter-20080625-2008.

The dynamically updated model parameters in 2011, 2013 and 2015 are different from each other, so are from parameter-20080625-2008, which is called parameter-

20080625-2008-updated. With these parameters, 8 flood events (parameters for the flood events in 2008 and 2009 are not updated) are simulated again, the four evaluation indices have been calculated and listed in Table 5, the simulated hydrographs are shown in Fig. 7 8 also to make comparison with those results simulated with no parameter updating.

                    Table 5 is here

From the results shown in Fig. 7 8 and Tabel 5, it has been found that the model performance has been improved with dynamic parameter updating. For example, for all the 8 flood events, the simulated hydrographs fit those of the observations better than

those simulated with no parameter updating. The Nash-Sutcliffe coefficients of all the simulated flood events with updated parameters gets higher, except those of flood event no. 20150520 and 20150720. The average Nash-Sutcliffe coefficient increasing is 0.764, a 0.3% increasing. While for the peak flow error, all flood events have observed decreasing, the average decreasing is 66.81%, a very significant model performance improrvement.

The above se results imply that with dynamical parameter updating, Liuxihe model has a much better performance in simulating the flood events of SRW, i.e., model parameters are in dynamic changing with the LUC changing, and dynamical parameter updating with the LUC changing is needed. This confirms the parameter dynamics assumption proposed in this study, and prove that the parameter updating method proposed in this study is effective. characteristics of model parameters.

**5 Discussions**

**54.13 Effect of parameter optimization on model performance**

To assess the efficacy of parameter optimization, simulations for the 12 flood events were conducted using the initial parameters, and the corresponding results are presented in Fig. 9. For comparative purposes, hydrographs obtained through simulations with dynamic parameter updating are also included. To test the effectiveness of parameter optimization, the 12 flood events are simulated with the initial parameters, and the results are shown in Fig 8, to make comparision, the simulated hydrographs with dynamically updated parameters are also shown. From the results, it could be found that the simulation results with initial and dynamic parameter updating dynamically updated parameters are quite different. Though both the simulated hydrographs have similar patterns, but the flows simulated with the initial parameters are generally much lower than the observations, and those simulated with dynamic parameter updating the dynamically updated parameters well fit the observation well.

Fig. 8 9 is here

The four evaluation indices of the 12 flood events is calculated and listed in Table 6. Compared with the simulation with initial parameter, the simulation with dynamic parameter updating dynamically updated parameters has been improved much based on these evaluation indices. Among them, the average Nash-Sutcliffe coefficient increased 68.2%, correlation coefficient increased 3.2%, peak flow error reduced 86.4%, water balance coefficient increased 45.8%. These results show that parameter optimization is needed and feasible even for distributed hydrological model.

Table 6 is here

**5 4.2 4 Flood simulation for Pparameter stationary**

In above sections, the dynamic parameter updating was based on the optimized parameter look-up tables with LUCs in 2018 2008. There appears a question, should the look-up tables be optimized with the latest LUCs, not the LUCs at a specific time? I.e., is are the look-up tables no-stationary to LUCs during urbanization. If yes, then the look-up tables needs to be optimized with the latest LUCs, and done again when there is significant LUCs change. Otherwise, the look-up tables can be optimized with LUCs at any given time, alleviating the need for frequent re-optimizationtime, and there is no need to optimize it very often. To answer this question, in this study, the parameters were optimized with LUCs in 2011, 2013 and 2015 , and the hydrological data used for parameter optimization were from flood events 20110516, 20130815 and 20150520

respectively. T, these parameters are called parameter-20110516-2011, parameter-

20130815-2013, and parameter-20150520-2015 respectively. Then the parameters are dynamically updated with latest LUCs, and are called parameter-20110516-2011- updated and parameter-20130815-2013-updated respectively. Notable, there is not parameter-20150520-2015-updated. In other words, dynamic parameter updating process in a forward temporal manner and not backward

. For example, parameter-20110516-2011- updated only update parameters with LUCs in 2013 and 2015, not in 2008 and 2011; parameter-20130815-2013-updated only update parameters with LUCs in 2015, not in

2008, 2011 and 2013, so there is not parameter-20150520-2015-updated. The dynamically optimized and updated parameters are then employed to simulate the flood events, and the results are shown in Fig. 10. The four evaluation indices are calculated and listed in table 7.

                          Table 7 is here

                          Fig. 10 is here

The simulated flood hydrographs, both with and without dynamic parameter optimization and updating, exhibit no significant differences. Drawing from the methods and results presented above, it can be concluded that the parameters demonstrate LUC stationarity during urbanization progress.

that the parameters are stationary during the urbanization, i.e., during the LUC changing period. There is no need to optimize the look-up tables frequentlyvery often with rapid

LUC changing. Instead, the focus should be on optimizing and updating of parameters, which emerges as the crucial aspect in response to dynamic LUCs conditions., parameter optimization and updating is most important.

**54.3 5 Impact of LUC changes on flood responses**

Based on the above results, it could be found that with due to the LUCs changes, the flood response changes also. To quantitatively analysis this effect, the peak flow and urban land area rate of flood events from 2011 to 2015 are extracted from the above results and listed in Table 8. The values with no-update are the simulated values with parameter-20080625-2008, while the ones with update are the simulated values with parameter-20080625-2008-update.

Table 8 is here

From the results, it could be found that from 2008 to 2011, the SRW observed an urban land rate change from 18.62% to 23.4%, a 25.67% increasing. For flood event

20110516, with the same precipitation, the peak flow will change from 87.08 $m^3$/s to

99.42 $m^3$/s, having with a 14.2% increasing. While for flood event 201100808, the peak flow changinge is from 103.68 $m^3$/s to 117.21 $m^3$/s, having a 13.1% increasing. Both these events are light flood, the peak flow increasing has similar magnitude.

But from 2008 to 2013, the SRW observed an urban land rate increasing of 40.92%.

For flood event 20130815, which is regarded as a heavy flood event, the peak flow increasing is 9.0%, while for flood event 20140511, the peak flow increasing is 12.8%, for flood event 20140819, this is 14.6%. The latter two flood events are regarded as
medium. With these results, it can be concluded that the much heavier of the flood
magnitude, the  less increasing of peak flow.
From 2008 to 2015, the SRW observed an urban land rate increasing of 63.10%. For
flood event 20150520, which is regarded as a light flood event, and flood events
20150523 and 20150720, which are regarded as a medium flood event, the peak flow
increasing are 56.3%, 18.5%, and 12.2% respectively. This implies that for the light
flood event, the peak flow increases much more. Based on this analysis, with the
increasing of the urban land area rate, the peak flow of a flood event will increase, and
the light flood event has the most peak flow increasing, while the heavy one has the
least peak flow increasing.

##  5 Conclusions

In this study, a method is proposed for accurately simulating flood processes of
urbanizing watersheds that appear during the world urbanization process, which
employs the Liuxihe model, a physically based distributed hydrological model as the
flood simulation tool. This method first derives initial parameter look-up tables, and
then optimizes them, and dynamically updates the parameters with the changing LUCs
to improve the model performance. A case study has been carried out in the
Songmushan Reservoir Watershed, a highly urbanizing watershed in the Pearl River
Delta Area in southern China which experienced rapid urbanization in the past decade.
Based on the results, following conclusions have been proposed.
1. The methodology proposed in this study could be used for simulating and forecasting
urbanizing watershed flood processes with good model performance.

2. For an urbanizing watershed, terrain properties are in changing, and model parameters are in changing also due to terrain properties changing, this is called model parameter dynamics. Model parameters should be updated with the LUC

changes.

3. Parameter look-up tables of physically  based distributed hydrological model are LUCs stationary, i.e., the parameter look-up tables only need to be determined once during the watershed urbanization.

4. With same precipitation, flood peak flow will increase due to urban land rate increases. The much heavier the precipitation, the less increasing of the peak flow.

5. This study provides more evidence to prove that parameter optimization is effective and needed in controlling parameter uncertainty for physically based distributed hydrological model.

**Competing interests:** The contact author has declared that none of the authors has any competing interests.

---

## Author Comment (AC2)

RC2: 'Comment on hess-2023-233'

General Comments:

Thank you for giving me the opportunity to review your paper. From my understanding, the main aspect of this paper is "Dynamic Parameter Updating." The method uses different parameter values over time based on the changing land cover. The parameter values are updated from the same look-up table at each flood event based on the land cover present at the event.

I recommend rejection as I do not believe the methods shown are novel enough to solicit publication in HESS. I believe that comparing Particle Swarm with other algorithms has already been done many times. I recommend going beyond a simple case study to better prove your lookup values are applicable to other urban catchments, or by adding more algorithms to compare against.

General Reply:

Dear Reviewer, thank you very much for your efforts to review our paper. We are very sorry that you do not give our paper a positive comment. We would like to mention that comparing Particle Swarm with other algorithms is not the purpose of this study, we only employed this algorithm to optimize the model parameter. The novelty of our paper is that we found that the model parameters of distributed hydrolgocial model is in changing with land use/cover change, i.e., the model parameter has dynamic characteristics. And we have proven this dynamic's existance, and the model parameter could be updated with the changing land use/cover. And we also find that the model parameters are LUCs stationary. Based on the literature review, no this kind of study has been published. We think the findings are novel and has its values.

We revised our paper based on your comments and another reviewer's comments, following are point-by-point responses to your comments. We finally hope you could change your mind. But unfortunately the revised manuscript could not be uploaded due to the Jounal's rule, but in the reply to another reviewer, we attached the major revision parts.

Many thanks.

Yangbo Chen, on behalf of all authors

Major Comments:

R1-C1. Can the figures be added in line with more descriptive captions? This seems minor but can deter readers from wanting to read your content if they cannot find your figures easily. I had to open two tabs when reviewing this, one for figures and one for the text, as I didn't want to lose my spot when reading.

AC1:Thank you for your reminder. As it is in the reviewing stage, so the figures are not put in the appropriate location. But after it is accepted, this problem should not exist anymore. Thank you for your suggestion, we revised Figure 7 (now Figure 8), add some explanatory words, so it is more readable.

[Figure]

(a) The evolution of parameters. Most parameters are assigned good initial value, except soil porosity coefficient, but it converged to the optimal value.

(b) The evolution of objective function. In this study, the objective function is to minimize the Mean relative error.

(c) The evolution of statistical indicators. 5 indices are employed.

(d) Simulation (20080625). Simulation result with optimized parameters and and the observed hydrograph.

Fig. 7 results of parameter optimization

R1-C2. Line 282-283: Is there a citation for the referring look-up values?

AC2:Thank you very much, dear reviewer. Two reference is cited (Chen et al, 2011; Chen et al, 2016), which is already in the reference list.

R1-C3. Lines 293-296. Can you better describe the initial values and the process that the Liuxihe model uses to attain them? This would benefit the reader as you refer to

these metrics heavily in your paper, say they are, "experience-based," but don't provide a method to calculate them/reproduce them.

AC3:Thank you very much, dear reviewer. In the references, there are detailed descriptions on how to determine these values. To avoid duplication, Line 281-291 already briefyly described this issue.

R1-C4. Line 377: Is there a citation on why PSO is effective with getting parameters and is "proven effective" when only using one flood event? Given the small amount of flood events used or parameter calibration, there is a chance that parameters are overfit

AC4:Thank you very much, dear reviewer. PSO is proposed for optimizing Liuxihe model parameters in 2016(Chen, Y., J. Li, and H. Xu, 2016. Improving flood forecasting capability of physically based distributed hydrological model by parameter optimization. Hydrol. Earth Syst. Sci., 20, 375-392. https://doi.org/10.5194/hess-20-375-2016). Then, dozens of studies have proven that hydrological data from only one flood event is needed to optimize the model parameters, which is different from the lumped hydrological model. There is no finding that overfit exists as the simulation to all other observed flood events are samely well. There is no problem of overfitting, the main reason is that distributed hydrological model has physical meaning, including the rainfall-runoff production and routing process, and the parameters.

R1-C5. Line 430, 436: Can there be a simpler flood naming convention? Using a long name with little formatting, parameter-20080625-2008, seems complex. Perhaps calling it Parameter set A, and then have a look-up table that maps your set letter to your dates. Using parameter-20080625-2008-updated could be called Parameter set B.

AC5:Dear reviewer, many thanks for your suggestion. This will cause new problem, i.e., the parameter set has no implicit connection with the flood event, which is emphasized in this study. For this reason, after a carefull consideration, we think it is better to maintain its current format. But anyway, think you very much for this suggestion.

R1-C6. The following comments are for Figure 6: a. For panel (a), what is the adjusting factor? b. For panel (b), what is your objective function? This is not described in the paper. c. For panel (c), it looks like your evaluation metrics (R, NSE, etc) start extremely close to their final values (after evolution 50) d. For panel (d), Can you provide a simulation with initial parameters as well? Based on the slight changes in metrics over all 50 epochs, my hypothesis is the initial parameters hydrograph would look very similar since your metrics in panel (c ) didn't vary much. A suggestion would be to compare the percent error of the initial simulation vs observations and the optimal parameter simulation vs observations. If your routing timing, and flood matching is more stable (closer to 0) after PSO, you can state that your optimal solutions are of a good fit. e. Line 391-394: You stated that because there was good fitting, the PSO algorithm worked. Can you add a quantitative metric/comparison with initial values to better show this?

AC6:Many thanks dear reviewer for your carefull review, below are point-by-point replies. Now Figure 6 becomes Figure 7 in the revised manuscript as shown above.

a. Adjusting factor is a coefficient adjusting the initial parameter value. During the optimization process, this factor changes, and finally reaches the optimal value. Simply, it is the ratio of the optimized parameter value with its initial value. For different parameter, this factor is different.

b. The objective function employed in this study is minimizing the Mean relative error.

c. Yes, you can say so. The reason is that after many year experiences in Liuxihe model parameterizaton, we have very good skill in determining the initial parameters. But there is exception in this study, initial value of the parameter of soil porosity coefficient is not so good, but finally with the optimization, the optimal value is found.

d. Please see Figure 8 (now Figure 9), there are comparisons among observation, simulation with initial parameter, and optimized parameters.

e. Please see "4.3 Effect of parameter optimization on model performance", and Figure 8 (now Figure 9), table 6.

R1-C7. Line 461: Are the initial parameters the ones used in Section 4.1, or the parameters generated from PSO at parameter-20080625-2008?

AC7:Yes, the initial parameters are the ones used in Section 4.1.

R1-C8. Figure 8: Can this figure be reworked? The text is very small/un-readable in the middle figures, and the precipitation varying with time change is hard to see.

AC8:Figure 8 (now Figure 9) is clear shown in my computer, so it is not redrawn. But if this paper could be accepted and published in HESS, and the editor find it is also needed to be redrwan, we certainly can do it.

R1-C9. For the conclusions, I'm not convinced by Conclusions 3, and 5. There was no experiment outlined for how the PSO algorithm was run, and if the data was overfit or not (only results were shown in Figure 6). Furthermore, the initial lookup table is based on "current experiences," which are not mentioned in this paper. Conclusions 3 is obvious based on the results of this study. Conclusion 5 has been proposed in the PSO reference and several applications, in this study, it is only be strenthened, particularly in the urbanizing watershed.

AC9:PSO algorithm for Liuxihe model flood modeling has been published in HESS in 2016 (Chen, Y., J. Li, and H. Xu, 2016. Improving flood forecasting capability of physically based distributed hydrological model by parameter optimization. Hydrol. Earth Syst. Sci., 20, 375-392. https://doi.org/10.5194/hess-20-375-2016), there are detailed description to this algorithm and its robustness has been proven in this paper, it is not the purpose of this paper to do it again.

The initial lookup table is based on the land use/cover type, it does not matter if it is based on "current experiences" or "past experiences", as they will be optimized, this does not change conclusion 3.

Conclusions 3 is obvious based on the results of this study, it is the purpose of this study. We think conclusion 3 is in no doubt.

PSO algorithm is not the key topic of this study. To make it clear, we revised conclusion 5 to as "This study provides more cluse to prove that parameter optimization is effective and needed in controlling parameter uncertainty for physically based distributed hydrological model.", or it may also be removed.

Minor Comments:

R1-C10. Line 163: The number 6 should be written out as a word, six.

AC10:Good comment, changed, thanks.

R1-C11. Figures 2, 4: Is there a better way to highlight the LUC change without having four similar basin plots? I can't really tell the difference between any of the yearly changes.

AC11:Good comment. Yes it is not easy to recongnize the difference by eye, it is for this reason, we add text following Figures 2 (Line 180-183), point out that the urban land area in 2008, 2011, 2013 and 2015 respectively, as urban land area is the most significant land use type.

For Figure 3, in Line 185-194, highest, lowest and mean elevation are listed, and 4 soil types percentages are also listed. These are good supplement to the information of the Figures.

For further information of Figure 4, line 523-534 provides more information.

We think these are good options, and some important information are presented clearly.

R1-C12. Line 306, 376: You already defined PSO, no need to redefine it.

AC12:Good comment, changed, thanks.

R1-C13. Line 376: There needs to be a space inserted between Optimization and the (

AC13:(PSO) deleted based on your previous comment, so this issue does not exist anymore, but anyway, thank you.

R1-C14. Line 376: It's Particle Swarm, not Particle Swam. This needs to be corrected in a few places.

AC14:Good comment, changed, many thanks.

R1-C15. Line 376: What algorithm/coding package was used for PSO?

AC15:no coding package, but Java is employed to write all the original code by ourself, which is embedded in the Liuxihe model software.